# Properties of Light Cementitious Composite Materials with Waste Wood Chips

**DOI:** 10.3390/ma15238669

**Published:** 2022-12-05

**Authors:** Huijuan Guo, Peihan Wang, Qiuyi Li, Guoying Liu, Qichang Fan, Gongbing Yue, Shuo Song, Shidong Zheng, Liang Wang, Yuanxin Guo

**Affiliations:** 1School of Architectural Engineering, Qingdao Agricultural University, Qingdao 266109, China; 2School of Civil Engineering, Qingdao University of Technology, Qingdao 266033, China; 3Shandong Junhong Environmental Technology Co., Ltd., Zibo 255000, China

**Keywords:** wood chip, cementitious materials, thermal conductivity, mechanical properties, pore structure

## Abstract

The CO_2_ emissions from the cement industry and the production of waste wood chips are increasing with the rapid growth of the construction industry. In order to develop a green environmental protection building material with low thermal conductivity and up to standard mechanical properties, in this study, pine waste wood chips were mixed into cement-based materials as fine aggregate, and three different kinds of cementitious binders were used, including sulfur aluminate cement (SAC), ordinary Portland cement (OPC), and granulated blast furnace slag (GBFS), to prepare a recycled light cementitious composite material. The mechanical, thermal conductivity, shrinkage, water absorption, and pore structure of a wood chip light cementitious composite material were studied by changing the Ch/B (the mass ratio of wood chip to binder). The results showed that the strength, dry density, and thermal conductivity of the specimens decreased significantly with the increase in the Ch/B, while the shrinkage, water absorption, and pore size increased with the increase in the Ch/B. By comparing three different kinds of cementitious binders, the dry density of the material prepared with OPC was 942 kg/m^3^, the compressive strength of the material prepared with SAC was 13.5 MPa, and the thermal conductivity of the material prepared with slag was the lowest at 0.15 W/m/K. From the perspective of low-cost and low-carbon emissions, it was determined that the best way to prepare a light cementitious composite with waste wood chips is to use granulated blast furnace slag (GBFS) as the cementitious binder.

## 1. Introduction

With the development of the construction industry in recent years, the amount of concrete and wood has risen sharply, consuming resources while also producing a large amount of wood waste [1,2]. Therefore, how to realize the reuse of wood waste to achieve the effect of reducing environmental pollution and resource waste has become an important research direction for the construction industry.

According to statistics, cement production is resource intensive and accounts for roughly 7% of the world’s total CO_2_ emissions [3,4,5]. Low energy consumption and sustainable development have become two globally relevant topics. Several scholars have incorporated wood waste into cement composites with the aim of developing new sustainable building materials with good performance and cost effectiveness. Several alternative wood–cement composites have been developed using wood as a filler [6] and wood waste ash as a partial replacement of cement [7,8]. Chowdhury et al. [9] used the microaggregate filling effect and volcanic ash activity of wood ash to achieve good results in the application of replacement cement. This will help to reduce the excessive consumption of cement and the impact of CO_2_ emissions during cement production on global warming.

In addition, the use of wood chips as an eco-aggregate to prepare lightweight concrete or mortar to reduce the weight of buildings while improving insulation performance is also becoming a growing topic of research [10]. Dias et al. [11] developed several composites with different wood chip contents to evaluate the potential of wood waste to replace conventional aggregates in concrete. The experiments showed that the compressive strength of the concrete decreased with an increasing wood chip content. Hafidh et al. [12] and other scholars [13,14] have also reported similar findings. Fadiel et al. [15] prepared wood concrete by replacing fine aggregate by volume with wood chips at different replacement levels of 5%~50% and proposed a formula for estimating the compressive strength based on the wood chip content. Batool et al. [16] emphasized the influence of sawdust as a fine aggregate on the hardening property of concrete and believed that a 10% substitution rate could obtain a low-cost composite with good performance. AlShuhail et al. [17] studied the effect of sawdust on the mechanical and physical properties of soil brick samples. The compressive strength increased to 1.72 MPa by adding 5% wood chips. Priya et al. [18] used 10%, 20%, and 30% blast furnace slag instead of ordinary Portland cement to prepare wood chip mortar, and the results showed that a mixture of 10% blast furnace slag and 5% sawdust resulted in a material with a better thermal insulation effect and mechanical properties close to 40 MPa. Ping et al. [19] revealed that the properties of wood chips on the processability, density, mechanical strength, and drying shrinkage of composites prepared from fly ash geopolymers were evaluated to broaden the application of wood chip in mineral admixtures. In addition, Qiuni et al. [20] studied the bending properties of five kinds of wood concrete composite decks bonded by epoxy resin or polyurethane. It is feasible to use wood chip concrete and engineering wood together as a composite structural plate, which has the ability to meet the requirements of bending strength and stiffness.

Some organic matter of the wood chips inhibits the compatibility of the substrate, resulting in an insufficient interfacial bonding between the substrate and wood chips, which in turn weakens the strength. Aiming to improve the interfacial bonding, the wood chips need to be pretreated; currently, the most common method is alkali treatment [21,22,23]. Tong et al. [24] pretreated wood chips with hot water, cold water, and alkaline solution. The results showed that the alkaline-solution-treated wood chip showed a better bonding interface in the cementitious system. Pascale et al. [25] used sodium silicate solution to saturate wood chips and showed that the compressive strength of the wood chip specimens treated with sodium silicate solution was 40% higher than that of the water-saturated ones. The presence of calcium alumina on the surface of the wood chip was observed by SEM, confirming the improved adhesion between the wood chip and the cement matrix. Xiaoshan et al. [26] proposed a new treatment that takes advantage of the fast setting of sulfur aluminate cement to consolidate the wood fiber precipitates before they dissolve, reducing the weakening effect of the precipitates on the strength.

A wood chip cement-based composite, as an innovative composite material, can also provide some additional functions, such as heat insulation and sound insulation, thus making up for its reduced mechanical energy to a certain extent [27,28]. Michał et al. [29] emphasized the advantages of using wood chips and reeds in wood concrete to improve the thermal insulation and sound insulation of concrete and to reduce the dead weight, and they developed building materials with good insulation performance. Almir et al. [30] prepared lightweight concrete by compounding wood chip and water-treatment sludge with a thermal conductivity coefficient of 1.89 W/m/K, which as 23% lower than the thermal conductivity of conventional concrete. In a study by Faiza et al. [31], the results showed that the wood–cement-based composite had excellent water regulation capacity, and the dynamic simulation results showed that the material significantly reduced the total energy consumption of the building.

The previous literature on wood chip cementitious materials was reviewed, and the results showed that wood chips can replace part of the natural sand as a fine aggregate for the preparation of wood chip cementitious composites, which has important research significance. However, there are few studies on the preparation of lightweight wood chip cementitious composites by replacing 100% natural sand with wood chips; therefore, this experiment investigated the feasibility of preparing lightweight mortar from wood chips as a single fine aggregate without adding natural sand. This method is an innovative point of this experiment to further reduce natural sand mining. Without adding natural sand, lighter materials with a better thermal insulation effect can be obtained, which conform to the concepts of energy conservation and sustainable development.

In this study, the aim was to develop a kind of lightweight wood chip cement-based material without natural sand. With cement as the control group, the feasibility of using alkali-excited blast furnace slag in wood chip cement-based material is discussed, so that it can achieve the effect of wood chip waste utilization and industrial solid waste. The mechanical properties, thermal conductivity, water absorption, dry density, shrinkage, and pore structure were comprehensively evaluated, and the effects of a cementitious binder type and the wood chip ratio on wood chip lightweight cementitious composites were analyzed to develop a green construction material with lower thermal conductivity and a high mechanical property.

## 2. Experimental Program

### 2.1. Raw Materials

The cement used in this study was sulfur aluminate cement and ordinary Portland cement produced by Tangshan Polar Bear Cement Co., Ltd. (Tangshan, China), and Shanshui Cement Co., Ltd. (Qingdao, China), respectively. The density of the sulfur aluminate cement was 2.95 g/cm^3^ and that of ordinary Portland cement was 3.16 g/cm^3^. According to the national standard of GB/T18046-2017 [32], the grade S95 blast furnace slag with a density of 2.88 kg/m^3^ was used; its alkalinity coefficient was 0.94 and an activity coefficient of 41% was obtained in the test. The chemical and physical properties of these three cementitious materials are given in Table 1.

The wood chips used in this study were pine waste wood chips with a density of 0.38 g/cm^3^ and a size of 0.5–2 mm. Figure 1 shows the appearance of an original wood chip and the microstructure of the wood chip observed by SEM (scanning electron microscopy). 

The detailed mineral composition and element analysis of the wood chips are given in Table 2. Figure 2 shows the cumulative screening curves for OPC (ordinary Portland cement), SAC (sulfur aluminate cement), GBFS (granulated blast furnace slag), and pine wood chips. Sodium hydroxide was produced by Zhiyuan Chemical Reagent (Tianjin, China) Co., Ltd., with an AR analytical purity of 96%. The sodium silicate solution was supplied by Qingdao Baoze Chemical Co., Ltd., and its indexes are given in Table 3. The water reducing agent was a powder polycarboxylic acid water reducing agent with a water reduction rate of 23~25%, produced by Qingdao Qingjian New Material Group Co., Ltd. (Singapore).

### 2.2. Mix Proportion Design

With reference to the existing production process of cement-based plant fiber materials and the actual situation of the experiments [33,34], on the premise of ensuring the compatibility and mechanical strength of the test specimens, five groups of test specimens of three different cementitious binders with a Ch/B of 1/10, 1/9, 1/8, 1/7, and 1/6, where Ch/B means the proportion of wood chip/binder, were prepared to conduct a comparative analysis, and three test specimens were prepared with each type of binder. The content of the cementitious binder was set at 400 kg/m^3^. The mass ratio of the water reducer was 0.5% of the cementitious material. The cementitious binder of the sulfur aluminate cement, ordinary Portland cement, and granulated blast furnace slag were marked groups A, B, C, respectively, and the detailed mix proportion is shown in Table 4.

The alkaline exciter solution of group C was prepared from sodium hydroxide and sodium silicate solution, and the alkaline exciter solution with a modulus of 1 and a concentration of 3% on the mass percentage of binder was used in this test. The cementitious binder and the untreated wood chips were added together into the mortar mixing device and mixed at a low speed for 30 s. After, the water or alkaline exciter was added and mixed at a slow speed for 60 s. Then, the water reducing agent was added, and the mixing was continued at a high speed for 90 s. The mixture was mixed and loaded into the prepared test mold. To remove air bubbles from the fresh mix, all specimens were shaken with a shaker for 30 s. The samples were then covered with cling film to prevent moisture evaporation. After curing for 24 h, the specimens were demolded and placed under standard conditions of 20 ± 2 °C and 90 ± 5% relative humidity for further conditioning.

### 2.3. Experimental Methods

In this study, testing for the mechanical properties, dry density, thermal conductivity, dry shrinkage, water absorption, and pore structure of the specimens was conducted to study the influences of the Ch/B ratio and the different binder types on the cementitious composites. Meanwhile, a SEM test was also conducted on the wood chip cement-based specimens to observe their microstructure.

In order to clarify the basic mechanical properties of the wood chip-filled lightweight cementitious composites, the 40 × 40 × 160 mm specimens were prepared for mechanical property tests.

According to the standard of JCJ/T 70-2009 [35], compressive and flexural strength tests on the specimens were carried out using a pressure testing machine. After 28 d of curing, the specimens were placed in an oven at 60 °C, dried to a constant weight, and moved out. After the specimen was cooled to room temperature, the dry density of the specimen was obtained by weighing and calculating. For the thermal conductivity tests, 300 × 300 × 30 mm specimens were prepared for each type of mortar. After 28 d of wet curing, they were placed in an oven at 60 °C and dried to a constant weight. The dried specimens were placed in a thermal conductivity tester for thermal conductivity determination. For the water absorption test, cubic specimens of 70 mm were prepared and standardized for 28 days, and the specimens were put into an oven at 75 °C, with the duration set to 24 h. After drying, the specimens were placed in a water bath with a water surface height of 35 mm, and the water absorption rate was recorded for 24 h.

Generally speaking, the testing for shrinkage of the mortar was for 7, 14, 21, 28, 56, and 90 days. Considering the specificity of the waste wood chips, the drying shrinkage was measured every 24 h for the first 7 days. The expression for the dry shrinkage rate is shown in Equation (1).
(1)$$\;\varepsilon_{at}=\frac{L_{t}}{L}\;$$
where $\varepsilon_{at}$ is the drying shrinkage rate at t days; $L_{t}$ is the shrinkage value of the specimen at t days (mm); and L is the length of the specimen (mm).

The internal pore structure of the cementitious materials was analyzed generally by mercury piezometry [36], the gas adsorption method [37], and the NMR (nuclear magnetic resonance) [38] method. The results of the mercury-pressure process are prone to be influenced by the pressure of the mercury feed, while the NMR technique is able to test the pore structure characteristics in situ, nondestructively, and accurately. Therefore, in this study, a nuclear magnetic resonance imaging analyzer, provided by Suzhou Newmark Analytical Instruments, was used to detect the pore size distribution of each specimen with a resonance frequency of 23 MHz and a magnet strength of 0.5 T. Meanwhile, the bond property between the pine wood chips and the cementitious materials was observed using a JCJ-7500 scanning electron microscope at an accelerating voltage of 5 kV. The preparation and experimental procedure for the light wood chip cementitious material specimens is shown in Figure 3.

## 3. Results and Discussion

### 3.1. Mechanical Properties

The effects of the incorporation of the pine wood chips and the type of cementitious binder on the compressive and flexural strength of the specimens were investigated. The results of the 28 d compressive strength and flexural strength of the different groups are shown in Figure 4 and Figure 5.

From Figure 4, it can be seen that the compressive strength of all the mortars in the different groups decreased with the increase in the Ch/B. The main reason might be that the high water absorption of the wood chips weakened the efficient hydration reaction of the cementitious composites. When the Ch/B was certain, the relationship between the magnitude of the compressive strength and each group of specimens was A > C > B. Previous studies showed that the soluble sugars precipitated from wood chips could slow down the hydration reaction rate [39]. To reduce this negative effect, an early strength sulfur aluminate cement was used in group A. The strength results also confirmed that the sugar precipitated from wood chips had little effect on the hydration reaction of the sulfur aluminate cement, resulting in slightly higher compressive strengths than those in both groups B and C, which indicates that this type of binder affects the compressive strength. Compared to control group B, the higher compressive strength in group C was mainly due to the fact that the alkaline exciter used in group C made the surface of the wood chip rougher, which in turn improved the bonding with the cement. When Ch/B reached 1/10, the strength results of A and C only differed by 0.9 MPa, but the compressive strength of group B was approximately 50% of that of group A. The Ch/B changed from 1/10 to 1/6, and the compressive strength of each group of specimens changed as follows: 53%, 78%, and 66% for groups A, B, and C, respectively. Compared with the natural fine aggregates, the bond between the wood chips and the cement matrix was relatively weaker, and the increase in the content of the wood chips also led to an increase in the porosity, which reduced the compressive strength. As can be seen from Figure 5, the flexural strength of the three groups of specimens tested was found to obtain the best flexural strength of 6.3 MPa for the A group specimens prepared from alkali-excited blast furnace slag and a 1/10 replacement ratio of wood chips. The loss of flexural strength gradually increased with the increase in the Ch/B, of which the nature of the wood chips and their compatibility resulted in a weaker bond strength between the binder and the wood chips. In addition, the mechanical property results indicated that the wood chip lightweight cementitious materials were not suitable for load-bearing structures.

From Figure 6, the SEM of wood chip lightweight cementitious composites can be seen. The white circles in Figure 6 show the larger pores, and the red ones show the interfacial adhesion between the pine chips and the cementing adhesive. The cementitious composites with the alkali-activated slag of group C can be observed to have less porosity and a denser microstructure. In contrast, the cementitious composite paste with the OPC and SAC added was mainly distributed outside the wood chips, and there were many pores inside the wood chips with a looser structure and poor interface adhesion.

### 3.2. Dry Density

The dry density results of the specimens of each group are shown in Figure 7. It can be seen that the dry densities of the specimens of these three groups were negatively correlated with the Ch/B, and the specimens of group B with lower compressive strength had a lower dry density. When the Ch/B increased from 1/10 to 1/6, the dry density of groups A, B, and C decreased by 27%, 28%, and 30%, respectively.

The reason for the decline in the dry density was that with the increase in wood chips, the water requirement of the fresh mix increased, and the internal water was used for the hydration reaction and drying evaporation as the curing age increased. Therefore, the quality of the specimens constantly decreased. The higher the wood chip content, the more water is absorbed and lost by the specimen, and the greater the overall dry density of the specimen decreases. The dry density of the tested specimens was lower than that of ordinary mortar; thus, the wood chip lightweight cementitious material could be classified as lightweight mortar.

### 3.3. Thermal Conductivity

Thermal conductivity is an important characteristics of the thermal insulation properties of a material. The thermal conductivity of each group is shown in Figure 8. The thermal conductivity of the three cementitious composites ranged from 0.24 to 0.15 W/m/K. Compared with the results of these three groups, it can be revealed that the differences in the thermal conductivity were small for the same amount of wood chips. This indicates that the effect of the binder type on the thermal conductivity of the specimens was not significant in this study. From Figure 8, it can be seen that the thermal conductivity decreased with the increase in the Ch/B, and the addition of wood chips improved the thermal insulation performance. With the change in the Ch/B, the thermal conductivity of the C group decreased by 32% to a final thermal conductivity of 0.15 W/m/K. The contribution of the wood chips to the thermal properties was mainly attributed to the increase in its content, leading to a decrease in the density of the mixture and an increase in the total porosity. In addition, it can be seen from Figure 6 that the material had more pore structures inside, which had the smallest thermal conductivity among the three phases of solid, liquid, and gas, thus providing the thermal insulation property. All of the thermal conductivities for the different groups in this study were lower than that of conventional mortar. Therefore, the obtained cementitious material in this study can be used as a new ecofriendly construction material.

### 3.4. Water Absorption Rate

The water absorption rate of each group of specimens was tested for 24 h, and the results are shown in Figure 9a. The results show that the water absorption rate of each group of specimens increased with the increase in the Ch/B (Ratio of sawdust content to cementitious material). The order of the water absorption rate of these three groups, from the largest to smallest, was B, C, and A; the 24 h water absorption rate increased by 193%, 240%, and 219%, respectively. The higher the wood chip content, the more obvious the increment in its water absorption rate. Changes in the 24 h water absorption of the specimens were also recorded, and the 24 h water absorption change curves are shown in Figure 9b–d. As can be seen, the water absorption rate of each group of specimens increased with time, the water absorption rate was faster in the early stage, and the water absorption rate could reach approximately 80% of the total water absorption rate in 12 h. This was caused by the high water absorption of the wood chips, and the water requirement of the mixture increased with the wood chip content, resulting in the increase in the structural porosity.

### 3.5. Drying Shrinkage

The testing for the drying shrinkage strain of the mortar was carried out according to the national standard of JGJ/T70-2009. The drying shrinkage curves of the three groups of specimens are shown in Figure 10. The results show that the drying shrinkage value increased with the increase in the wood chip content. This was likely to due to the higher water absorption of the wood chips than the natural aggregate.

It can be seen from Figure 10 that the drying shrinkage of group B was the largest, followed by group C, and group A had minor shrinkage. The main reason was that the pore size of group A was the smallest, which slowed down the loss of free water. The pore size of group B was the largest, and a large amount of hydration heat was released during the cement hydration process, which accelerated the evaporation of water, leading to the largest drying shrinkage. Group C’s sawdust was mineralized after being treated with an alkaline activator and slag, which sped up the hydration process to a certain extent, resulting in more significant shrinkage than that of group A.

### 3.6. Pore Structure

The NMR technique was used to test the specimens with the size of 40 × 40 × 40 mm cured over 28 days. The transverse relaxation time T2 distribution of the specimens was obtained, as shown in Figure 11. The areas surrounded by the T2 spectrum curve and the *x*-axis characterize the internal pore area of the specimens, the transverse relaxation time was proportional to the pore diameter, and the change in the T2 spectrum can reflect the change in the internal pore structure of the specimen [40]. It can be seen that the T2 distribution of the binder SAC, slag (Figure 11a,c), and OPC (Figure 11b) were significantly different. The internal pores of the specimen with the SAC and slag as the binder were relatively small, and the signal peaks were mainly distributed in 0.1~10 ms, while the internal pores of the specimen with the OPC as the binder were relatively large, and the signal peaks were wider and mainly distributed in 10~1000 ms.

From Figure 11a–c, with the increase in the Ch/B ratio, the primary peak value of T2 moved downward to the right, and the aperture size gradually increased. When the Ch/B ratio was 1/10, the pore sizes of groups A and C were the smallest. When the Ch/B ratio was 1/6, the pore size reached the largest. The reason was that the pore diameter of the specimen became larger with the increase in the wood chip content. The pore size distribution was no longer uniform because the excessive wood chip could not be evenly distributed in the cement paste, and some wood chip agglomerates tangled together to produce weak parts, leading to loose structures and a lower compressive strength.

## 4. Conclusions

The study was conducted by comparing three binders, five kinds of Ch/B (wood-chip-to-binder-mass ratio), and the thermal conductivity and mechanical properties of the cement-based composites, and the following conclusions were obtained.

(1)The strength, dry density, and thermal conductivity of the composites decreased with the increase in the Ch/B ratio, while the water absorption, pore size, and drying shrinkage of the composites increased with the increase in the Ch/B ratio.(2)The thermal conductivity of the cementitious composite material was lower between 0.24 and 0.15 W/m/K than that of conventional mortar. The addition of wood chips improved the thermal insulation properties of the material and reduced the dry density, and the water absorption rate showed an increasing trend.(3)The T2 spectrum characterized that the increase in the wood chip content led to an increase in the pores in the specimens, and the pores of the specimens in the sulfur aluminate cement and blast furnace slag groups are mostly small pores, while the pore size of the ordinary Portland group was relatively larger.(4)When the Ch/B was 1/10, the strength of the sulfur aluminate cement group and blast furnace slag group was 98% and 85% higher than the ordinary Portland cement group, respectively. The drying shrinkage of the blast furnace slag group was lower than 4% of that of the ordinary Portland group, and the production process of the blast furnace slag was energetic, waste friendly, and environmentally friendly, and the group had good mechanical properties and thermal insulation properties. Therefore, the best performance of the wood chip lightweight cement matrix composites prepared with alkali slag was considered as the best choice of binder.

## Figures and Tables

**Figure 1 materials-15-08669-f001:**
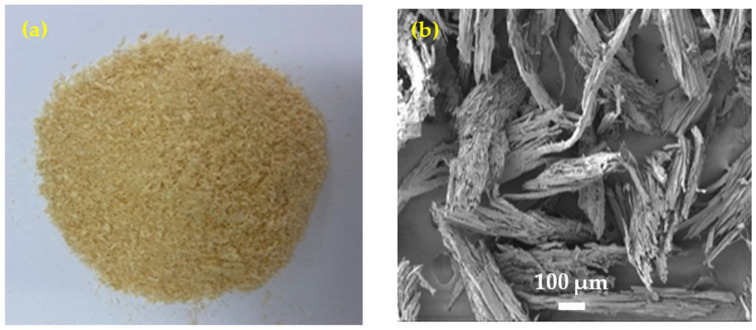
Wood chip morphology: (**a**) appearance of the wood chip; (**b**) SEM image of the wood chip’s microstructure.

**Figure 2 materials-15-08669-f002:**
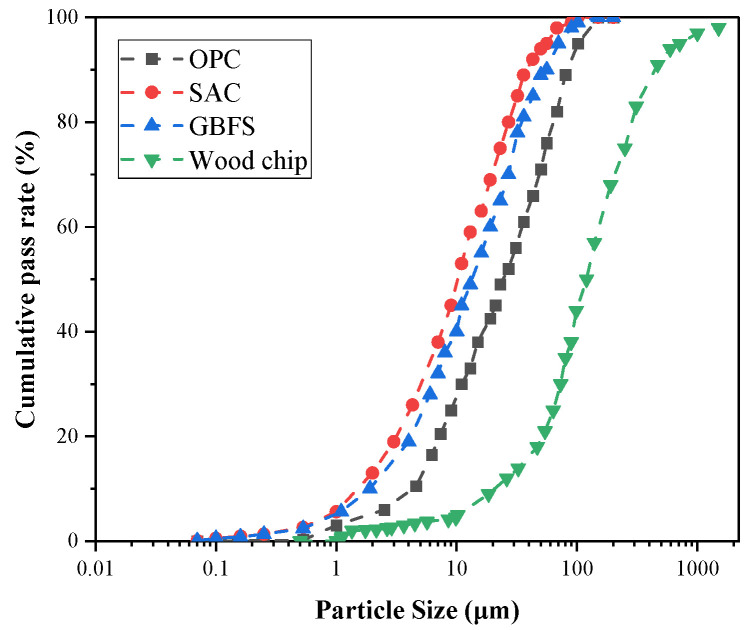
Cumulative particle size distribution curves of the pine wood chip and cementitious material.

**Figure 3 materials-15-08669-f003:**
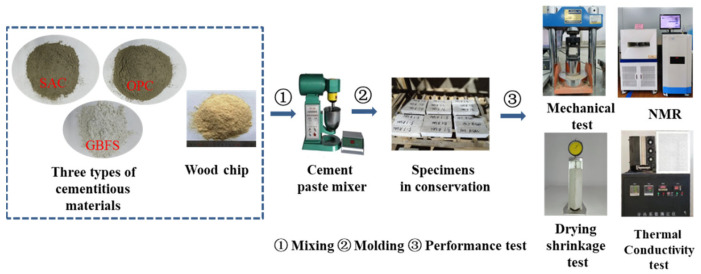
Preparation and experimental process of the specimens.

**Figure 4 materials-15-08669-f004:**
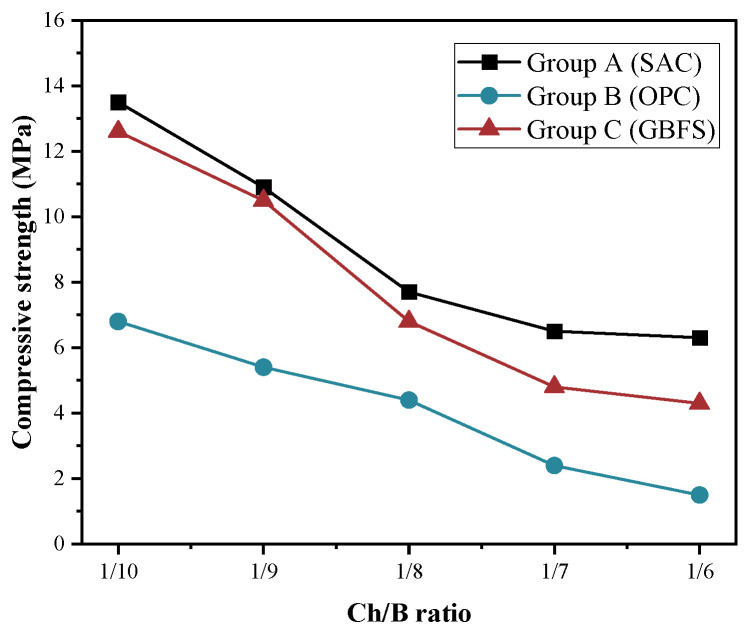
Compressive strength of the different series at 28 d.

**Figure 5 materials-15-08669-f005:**
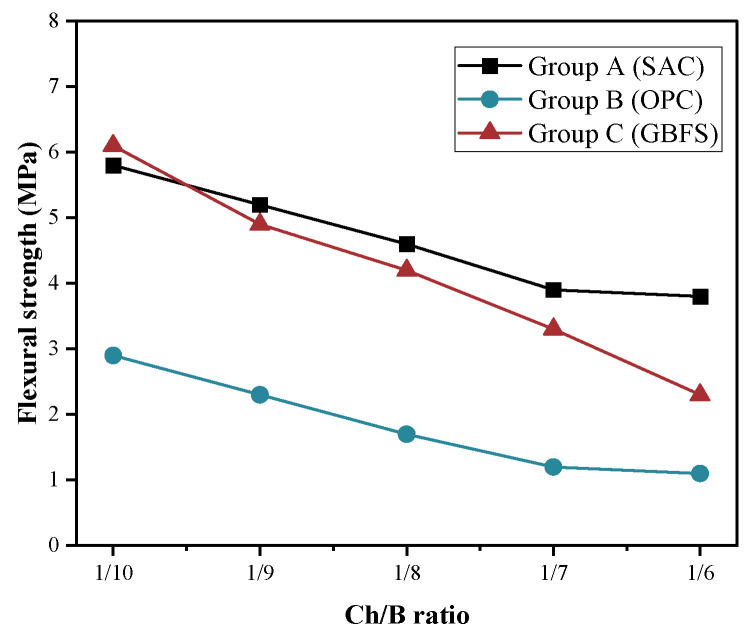
Flexural strength of the different series at 28 d.

**Figure 6 materials-15-08669-f006:**
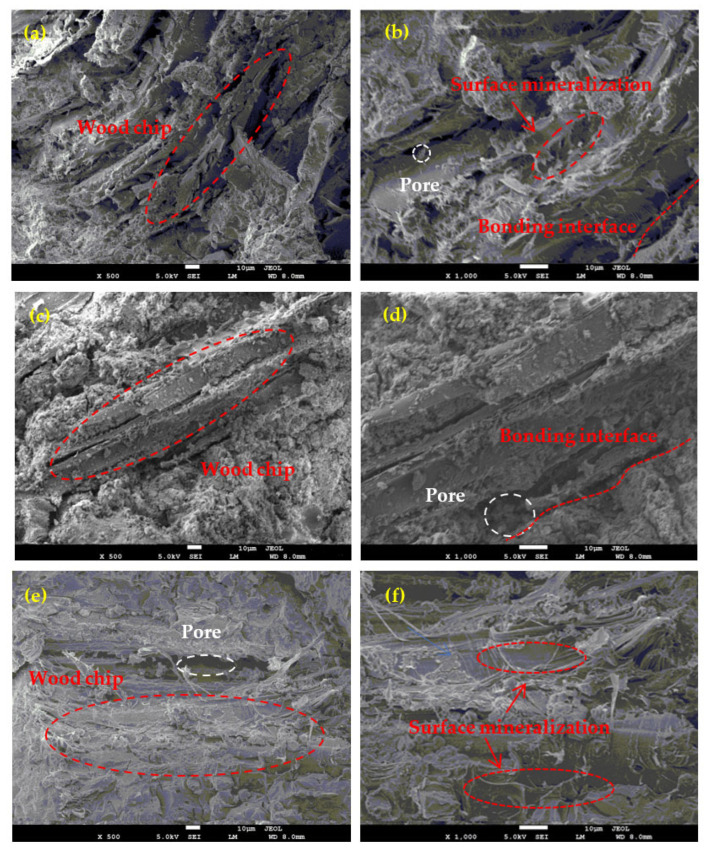
SEM of wood chip lightweight cementitious composites: (**a**) OPC 500 times; (**b**) OPC 1000 times; (**c**) SAC 500 times; (**d**) SAC 1000 times; (**e**) GBFS 500 times; (**f**) GBFS 1000 times.

**Figure 7 materials-15-08669-f007:**
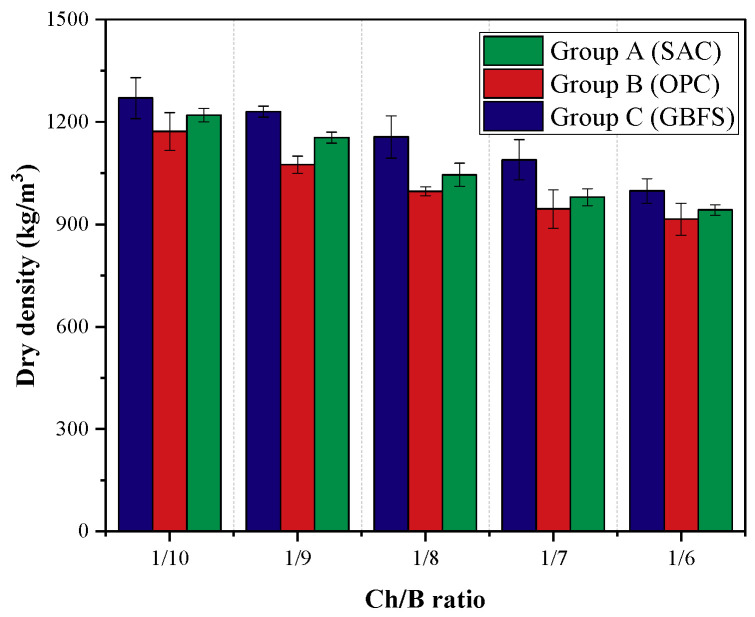
Dry density of the different groups.

**Figure 8 materials-15-08669-f008:**
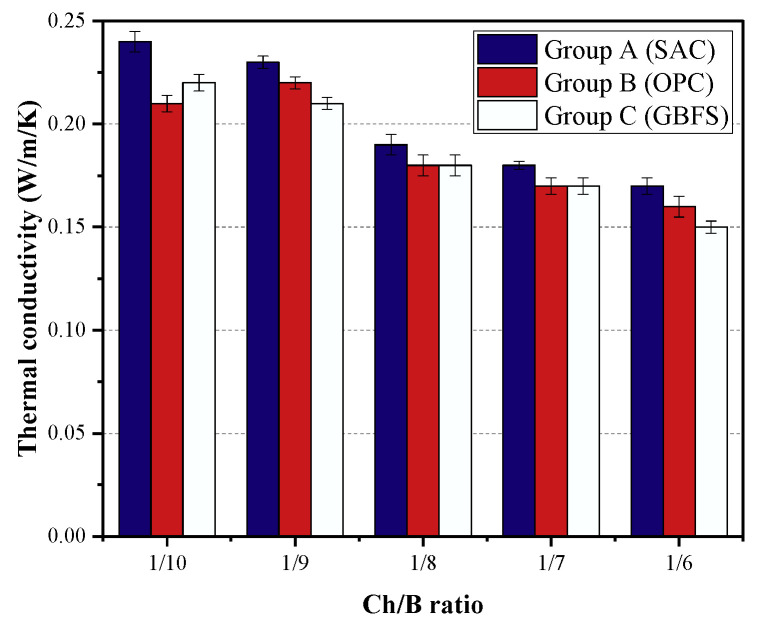
Thermal conductivity of the different groups.

**Figure 9 materials-15-08669-f009:**
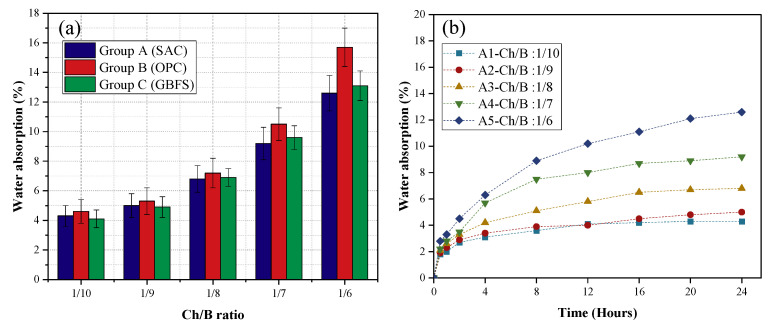
The results of the water absorption rate: (**a**) water absorption rate of the different groups over 24 h; (**b**) water absorption rate of the SAC specimen; (**c**) water absorption rate of the OPC specimen; (**d**) water absorption rate of the GBFS specimen.

**Figure 10 materials-15-08669-f010:**
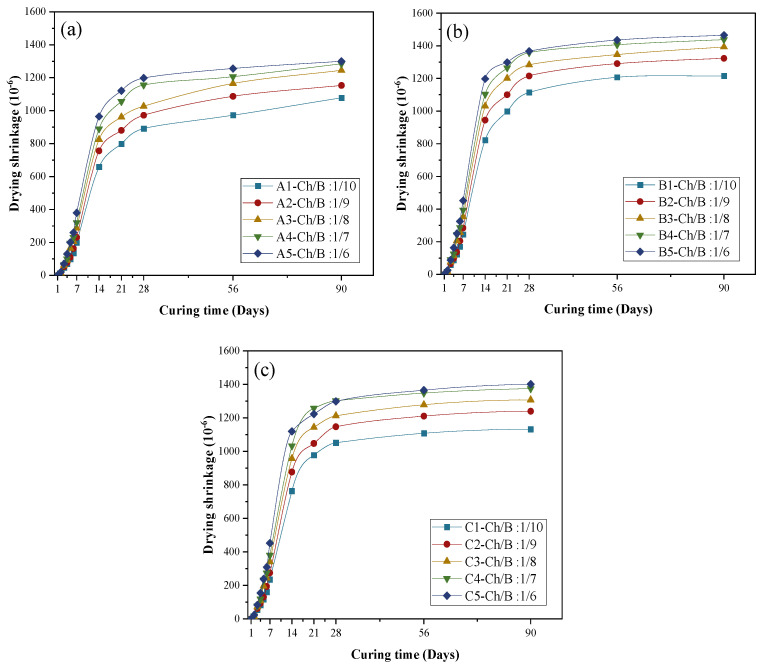
Drying shrinkage in the different groups: (**a**) drying shrinkage of the SAC specimen; (**b**) drying shrinkage of the OPC specimen; (**c**) drying shrinkage of the GBFS specimen.

**Figure 11 materials-15-08669-f011:**
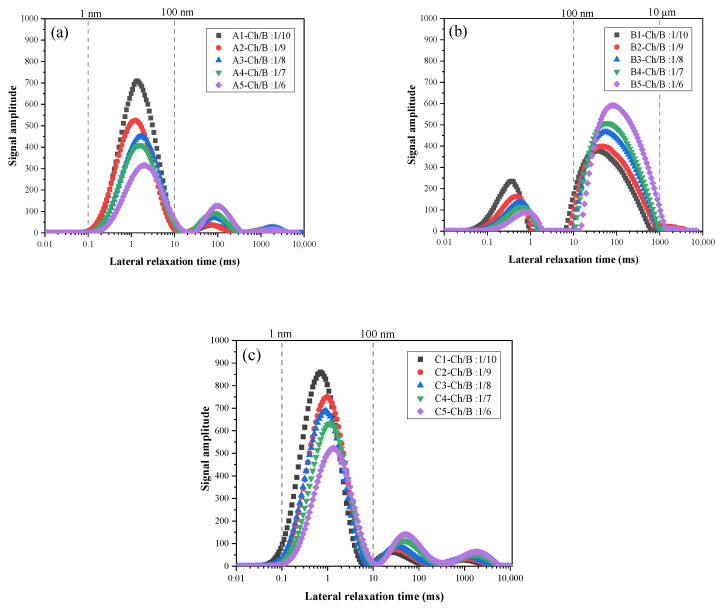
Variation of the T2 spectrum distribution of the different groups: (**a**) T2 spectrum of the SAC specimen; (**b**) T2 spectrum of the OPC specimen; (**c**) T2 spectrum of the GFBS specimen.

**Table 1 materials-15-08669-t001:** Chemical and physical properties of different cementitious composite materials.

Chemical Composition	SiO_2_ (%)	Al_2_O_3_ (%)	Fe_2_O_3_ (%)	CaO (%)	MgO (%)	SO_3_ (%)	Others (%)	Loss (%)	Specific Surface Area (kg/m^2^)
SAC	10.59	31.80	3.08	41.69	3.76	7.32	1.39	0.37	410
OPC	21.83	6.34	3.21	64.35	1.42	0.32	1.82	2.48	390
GFBS	18.2	4.7	0.94	64.6	9.5	0	2.72	1.58	430

**Table 2 materials-15-08669-t002:** Elemental analysis and chemical composition of the pine wood chips.

Elemental Analysis (%)			Chemical Composition (%)			Calorific Value (MJ·kg^−1^)
C	H	O	Lignin	Cellulose	Hemicellulose	18.5
48.20	6.08	45.40	22.4	41–46	30.2	

**Table 3 materials-15-08669-t003:** Physical and chemical parameters of the sodium silicate solution.

Na_2_O (%)	SiO_2_ (%)	Modulus (Ms)	Baume Degree (20 °C)	Density (g/mL)	Water Insoluble Matter (%)
8.74	28.16	3.3	39.8	1.37	0.01

**Table 4 materials-15-08669-t004:** Mix proportion design of the mortar.

Group	Ch/B Ratio	Binder Content (g)	Wood Chip Content (g)	Water (g)	Water Reducer (%)
Group A (SAC)	1/10	400	40	178	0.5
	1/9	400	44.4	188	0.5
	1/8	400	50	204	0.5
	1/7	400	57.1	227	0.5
	1/6	400	66.7	252	0.5
Group B (OPC)	1/10	400	40	177	0.5
	1/9	400	44.4	188	0.5
	1/8	400	50	204	0.5
	1/7	400	57.1	225	0.5
	1/6	400	66.7	253	0.5
Group C (GBFS)	1/10	400	40	142	0.5
	1/9	400	44.4	155	0.5
	1/8	400	50	173	0.5
	1/7	400	57.1	187	0.5
	1/6	400	66.7	242	0.5

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
