# Peer review of "Properties of Light Cementitious Composite Materials with Waste Wood Chips"

_materials, 2022, doi:10.3390/ma15238669_

Round 1

Reviewer 1 Report

The review comments are attached.

Author Response

Thanks for giving us the chance to revise our manuscript entitled “Properties of light cementitious composite materials with waste wood chips” (Manuscript Number: materials-2006683)

The manuscript has been carefully revised according to the reviewers’ comments and the itemized response is attached. We now submit our revised manuscript for your kind re-consideration.

Reviewer 2 Report

A comaparative economical benefit should be incorporated to improve the substance of the articel. Please see the marked manuscript for further details of comments.

Author Response

(The authors gave the same response as above.)

Reviewer 3 Report

Paper ID: materials-2006683

Type: Article 
Title: 
Study on properties of light cementitious composite materials with waste wood chips

Authors: Guo Hui juan , Wang Peihan , Li Qiuyi , Liu Guoying , Fan Qichang , Yue Gongbing , Song Shuo , Zheng Shidong , Guo Yuanxin , Wang Liang 

This study investigates properties of light cementitious composite materials with waste wood chips. Although the testing methods and compared results attained in the present study show the importance of the paper, The authors should address the following comments:

 Novelty in comparison to recent literature? Need to be emphasized.

  1. The results in the paper might be more discussed by the relevant literature.
  2. Abstract: Please use comma (,) instead of “;”.
  3. Lines 39-41. Please combine them.
  4. Lines 46-48: “S. Chowdhury et al. used the micro-aggregate filling effect and 46 volcanic ash activity of wood ash to achieve good results in the application of replacement cement [5-8].” References 5-8 totally different.
  5. Please corrected the reference style in the text. Exmp. Valeria et al. [11].
  6. I strongly suggest that the introduction section should be rewritten.
  7. There should be a space between number and unit. Please correct these errors in the paper.
  8. Throughout the text, there are some typos that must be eliminated.
  9. There should be a space between number and unit.
  10. I suggest that the particle size distributions of Cement, GBFS and SAC should be added.
  11. Fig 1: Please check a and b.
  12. Fig 1b: Please add line scale with high resolution.
  13.  Please use verb of “given” for Table and shown for “Figures”
  14. Please add error bars to Figures. Error bar of the figures can help to show the distinction between the samples.
  15. Fig 6: Please check a and b.

Author Response

(The authors gave the same response as above.)

Round 2

Reviewer 1 Report

The authors have addressed all my questions and I have no other comments.

Reviewer 3 Report

The authors have made the necessary changes. Therefore manuscript can be accepted as it is.